# Non-Destructive Characterization of Selected Types of Films and Other Layers via White Light Reflectance Spectroscopy (WLRS)

Dimitrios Goustouridis [1], Ioannis Raptis [1,2,*], Theodora Mpatzaka [1], Savvina Fournari [1], Grigorios Zisis [1], Panagiota Petrou [3] and Konstantinos G. Beltsios [4,*]

1 ThetaMetrisis S.A., 12132 Athens, Greece
2 Institute of Nanoscience and Nanotechnology, NCSR "Demokritos", 15310 Athens, Greece
3 Immunoassays/Immunosensors Lab, Institute of Nuclear & Radiological Sciences & Technology, Energy & Safety, NCSR "Demokritos", 15341 Athens, Greece
4 School of Chemical Engineering, National Technical University of Athens, Zographou Campus, Attiki, 15780 Athens, Greece
* Correspondence: raptis@thetametrisis.com (I.R.); kgbelt@mail.ntua.gr (K.G.B.)

**Abstract:** In this work, we consider White Light Reflectance Spectroscopy (WLRS) as an optical methodology for the accurate, fast and non-destructive measurement of film thickness in the 1 nm to the 1 mm range and for applications that include microelectronics, photonics, bioanalysis and packaging. Films to which WLRS is applicable can be either homogeneous or layered-composite ones, while thickness and composition might be fixed or varying with time; in the latter case, real-time monitoring of the kinetics of processes such as certain transitions, film dissolution and bioreactions is possible. We present the basic principles of WLRS and a selection of characteristic application examples of current interest, and we also briefly compare WLRS with alternative methods for film measurement.

**Keywords:** white light reflectance spectroscopy; ultra-thin films; ultra-thick films; graphene; glass transition of films; biomolecular interactions; transparent films; film dissolution

## 1. Introduction

Coatings are playing a crucial role in our lives and their impact on future technologies and products is expected to further increase; if nothing else, one category of popular nanomaterials is that of layers having nanoscale thickness. Therefore, it is important to produce these coatings in large quantities and at low cost, and at the same time accurately control their properties. The physical and chemical properties of each coating define the characteristics of the coating and thus define the areas where each coating can be applied. Currently, several technologies for the characterization of the properties of coatings [1–3] are mature and commercially available.

Thickness is amongst the fundamental properties of any coating and ranges from nanometer to millimeter scale. Quite often, the properties of a coating depend strongly on the thickness of the coating because of a different amount of material per substrate area and/or for different reasons, such as a coating structure that is thickness dependent or diminishes with increasing thickness, contributing to the coating–substrate interface to the property in consideration. It is then of paramount importance to use methodologies that can measure the coating thickness in a non-destructive and non-contact way and deliver accurate results within a very short time. Methodologies considered for the measurement of the thickness of a coating include stylus profilometry [4], eddy current [5], X-ray Reflectance (XRR) [6], ellipsometry [7], White Light Interferometry (WLI) [8] and White Light Reflectance Spectroscopy (WLRS) [9], the last also termed more briefly as Reflectance

Spectroscopy. These methodologies can be applied to different types of materials and for various thickness ranges. From these methodologies, the optical ones appear to present some unique advantages, such as non-destructive character of measurements and very high resolution. In certain cases, the calculation of additional important film properties such as the refractive index of the film(s) of interest is also possible.

Ellipsometry, either single wavelength [10] or spectroscopic [11], is a very powerful methodology that is exploited in several commercial products. It is the method of reference for ultra-thin films and is capable of measuring film thicknesses up to a few micrometers. However, the ellipsometric setups are rather bulky and their implementation with small spot-size is not straightforward.

Reflectance spectroscopy is based on the illumination of the sample under characterization with broadband light and the analysis of the reflected light with a high-resolution miniaturized spectrometer and further fitting with the appropriate models and fast algorithms. In this work, reflectance spectroscopy capabilities are exploited in a set of highly demanding and diverse applications, and its advantages against other applicable methodologies are discussed. In particular, we consider the issues of thickness (cases of ultra-thin and ultra-thick films) and transparency (transparent films and transparent substrates) and the probing of multi-layer films for cases involving film thickness changing with time as a result of dissolution, biomolecular interactions or a glass transition (monitored at a fixed heating rate). Overall, reflectance spectroscopy is a very powerful methodology for the fast, accurate, affordable and non-destructive measurement of film thickness of single or multi-layer stacks capable to serve a wide range of diverse applications.

## 2. WLRS Methodology

In Figure 1, the schematic of the setup for the characterization of coatings through reflectance spectroscopy is illustrated, considering the generic case of a single layer applied on a semi-transparent substrate, e.g., $SiO_2$/Si. The measurement setup consists of three main modules: the light source, the spectrometer and the reflection probe. The light from the light source is coupled to one branch of the reflection probe that is mounted over the sample under characterization and illuminated at 90°. As a result of different refractive indices, the light that is incident on the sample/air interface is partly reflected and the rest propagates in the sample under characterization; the same happens to the latter part of the light once it reaches the film/substrate interface. In the general case of thick and semi-transparent substrates, the transmitted light is absorbed. The reflected light from the film/substrate interface reaches the film/air interface and a significant percentage is transmitted to the air. The total reflected light is collected by the other branch of the reflection probe and is guided to the spectrometer for further analysis.

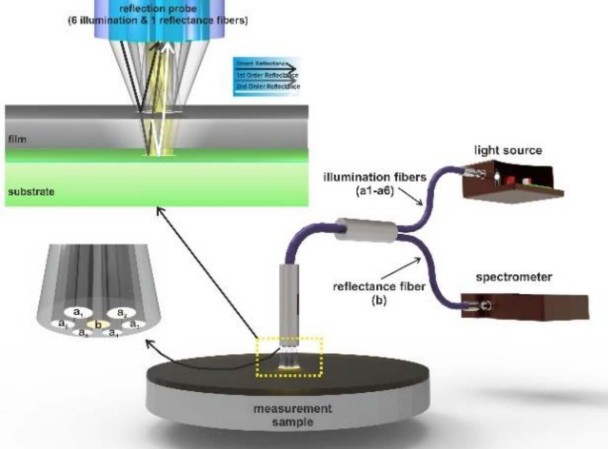

**Figure 1.** Typical WLRS setup: (a) detail of the reflection probe configuration; (b) detail of the light optical path through a single layer on semi-transparent substrate.

For the particular case of one transparent film on substrate, the reflectance signal vs. wavelength is:

$$R(\lambda) = \frac{r_{01}{}^2 + r_{12}{}^2 + 2r_{01}r_{12}\cos\left(\frac{4\pi n_1(\lambda)d}{\lambda}\right)}{1 + r_{01}{}^2 r_{12}{}^2 + 2r_{01}r_{12}\cos\left(\frac{4\pi n_1(\lambda)d}{\lambda}\right)} \tag{1}$$

and

$$r_{01} = \frac{n_0 - n_1}{n_0 + n_1} \tag{2a}$$

$$r_{12} = \frac{n_1 - n_2}{n_1 + n_2} \tag{2b}$$

where $\lambda$ is the wavelength; $r_{ij}$ are the relative refractive indices between adjacent layers (0 is considered as ambient, 1 the film and 2 the Si substrate); $n_0(\lambda)$, $n_1(\lambda)$ and $n_2(\lambda)$ are the refractive indices of the three layers; and $d$ is the thickness of the film [12].

In Figure 2, the theoretical reflectance spectra for three characteristic cases and a wide spectral range are illustrated: (a) 10 nm $SiO_2$ layer on Si, (b) 1000 nm $SiO_2$ on Si, (c) 100 μm $SiO_2$ on Si. In Figure 2a, the theoretical reflectance of the Si substrate without any coating is also illustrated for comparison purposes. The interference fringes are dense in the UV part of the spectrum and sparse in the NIR part. Thus, for the measurement of thick and ultra-thick films, the optical setup should be tuned to operate at long wavelengths and with the maximum possible optical resolution. On the other hand, for the measurement of thinner films, from a few hundreds of nanometers up to a few tens of microns, any part of the spectrum can be used for the measurement of thickness. However, for thin and ultra-thin films (e.g., <100 nm) interference fringes are not observed and in such cases the measurement of film thickness relies largely on the resolution and accuracy of the reflectance measurement in the UV part of the spectrum.

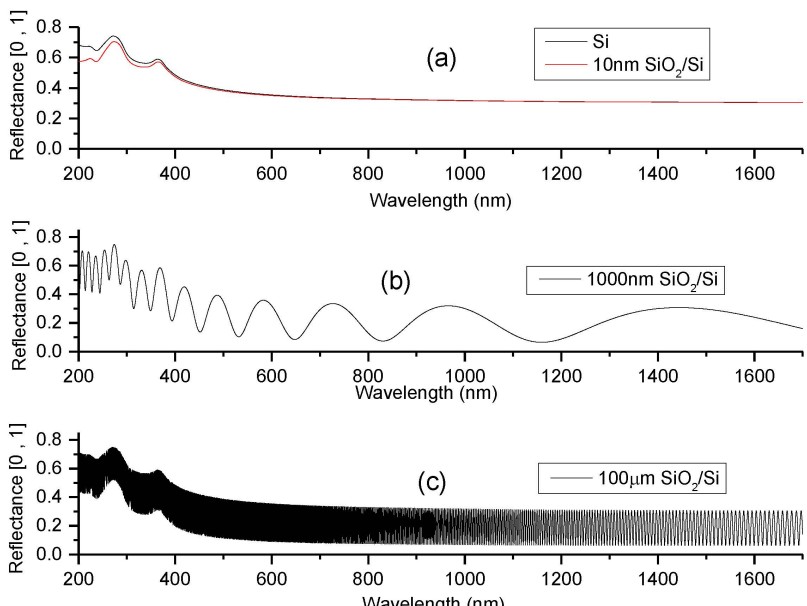

**Figure 2.** Theoretical reflectance spectra for various $SiO_2$ thicknesses on Si substrate: (**a**) plain Si substrate and 10 nm $SiO_2$/Si; (**b**) 1000 nm $SiO_2$/Si and (**c**) 100 μm $SiO_2$/Si.

## 3. Results and Discussion

We now proceed to the presentation of a range of micron scale to nanoscale applications of WLRS. We present and discuss our data for representative examples and emphasize fine details and occasional limitations. Issues pertinent to size (thickness) scale, transparency, refractive index difference, heterogeneities and real-time monitoring are addressed.

### 3.1. Characterization of Ultra-Thin Films

Thin and ultra-thin films play a pivotal role in numerous applications and in particular in high-end microelectronic devices. In these devices, the actual thickness is critical for the optimum operation of the device; consequently, the measurement of the thickness at nanometer level is of paramount importance [13,14].

In Figure 3a, the theoretical reflectance spectra of various $SiO_2$ films on Si substrate with $SiO_2$ thickness <200 nm are illustrated. In particular, the selected $SiO_2$ thickness values are 200, 100, 50, 20, 5 and 2 nm, while the reflectance from plain Si is also included. The same reflectance spectra are illustrated in Figure 3b but for the UV part of the spectrum (200–400 nm wavelength). Clearly, for film thickness < 20 nm, there is no interference extremum in the VIS/NIR part of the spectrum (400–1700 nm). Thus, the expected $SiO_2$ minimum thickness that could be measured in this spectral range is close to 20 nm. On the contrary, the UV reflectance spectrum from 20 nm $SiO_2$/Si is considerably different from the reflectance of Si (thick line), so this part of the spectrum should be employed in the measurement of thin films. The reflectance spectrum for all $SiO_2$ films with thickness <20 nm are of similar shape, with the exception of reflectance values. For example, the reflectance difference between plain Si and 2 nm $SiO_2$/Si is 1.2% at 200 nm wavelength and 0.1% at 400 nm wavelength, while the reflectance difference between plain Si and 5 nm $SiO_2$/Si is 6.3% at 200 nm wavelength and 0.7% at 400 nm wavelength. Therefore, the measurement of ultra-thin films (<10 nm thickness) necessitates very accurate measurement of the reflectance for both the reference material and the sample itself and same distance between the sample and the probe for both the reference and the sample under characterization.

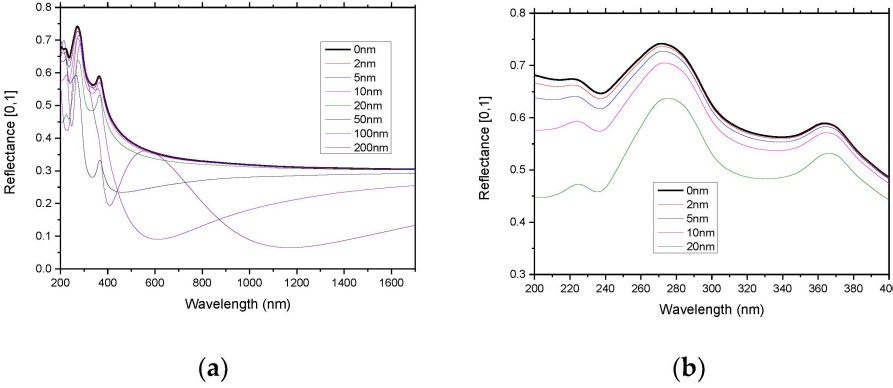

(**a**)                                   (**b**)

**Figure 3.** (**a**) Reflectance spectra of $SiO_2$/Si over the 200–1700 nm spectral range and for $SiO_2$ thickness in the 0–200 nm range. (**b**) Reflectance spectra of $SiO_2$/Si over the 200–400 nm spectral range and for $SiO_2$ thickness in the 0–20 nm range.

In Figure 4a, the experimental and fitted spectra (200–400 nm) of ultra-thin $SiO_2$/Si layers for $SiO_2$ thickness of 3.3 nm is illustrated. The $SiO_2$ thickness has been verified by spectroscopic ellipsometry. For thicker films, the fitting can be applied at higher wavelengths; for example, in Figure 4b the measurement of 17 nm $Si_3N_4$ is demonstrated through fitting in the 400–900 nm spectral range. Therefore, reflectance spectroscopy is capable of providing accurate characterization of ultra-thin dielectric layers grown on smooth semi-transparent substrates.

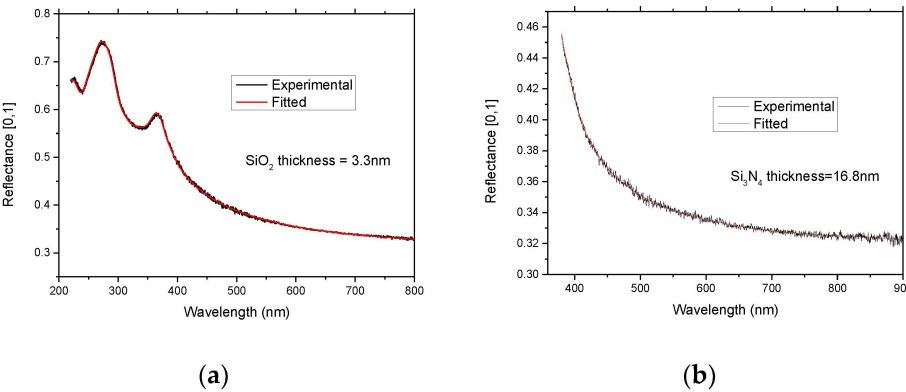

(**a**) (**b**)

**Figure 4.** Experimental and fitted reflectance spectra of thin dielectric films on semi-transparent substrates: (**a**) 3.3 nm $SiO_2$/Si and (**b**) 16.8 nm $Si_3N_4$/Si.

### 3.2. Characterization of 2D Materials

Two-dimensional materials have attracted significant attention during the recent years, because of their unique properties that can be exploited in a wide range of applications. One of the technological challenges that needs to be addressed is the controlled deposition of 2D materials for further processing. In Figure 5a, an optical microscope image (2.5×) of a graphene sheet deposited on $SiO_2$/Si substrate is illustrated. Owing to the refractive index contrast between graphene and $SiO_2$/Si, a free graphene area with a diameter of ca. 2.5 mm is clearly distinguished. In Figure 5b, the thickness measurement results for an $8 \times 8$ mm$^2$ area is presented. In the white area, only the $SiO_2$ layer was detected; in the brown area, a graphene layer with thickness 0.42 nm was measured. The graphene layer was also measured by spectroscopic ellipsometry and very good agreement was found.

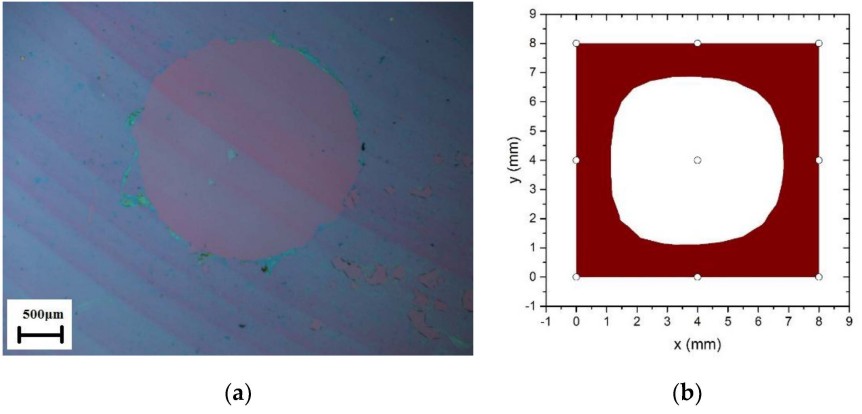

(**a**) (**b**)

**Figure 5.** (**a**) Image from an optical microscope (objective lens 2.5×) showing the graphene sheet deposited on $SiO_2$/Si. (**b**) Thickness mapping by reflectance spectroscopy over an area of $8 \times 8$ mm$^2$. Clearly, a circle with the same diameter without any graphene is measured (white area).

### 3.3. Characterization of Ultra-Thick Films

Thick and ultra-thick films are quite common for a wide range of applications, e.g., [15,16]. For the thickness measurement of thick and ultra-thick films, the fitting of the reflectance spectrum should be performed in NIR. In this application area, there are two strategies that could be applied: (a) use of spectrometers with a high number of pixels Si-CCDs tuned in the 900–1100 nm spectral range, and (b) use of spectrometers with InGaAs pixel arrays tuned to operate in a narrow spectral range, e.g., 1280 to 1350 nm.

The first approach is based on the use of Si-CCD-based spectrometers with 3600 pixels, which, when accompanied with narrow entrance slits, deliver an extremely fine optical resolution, even better than 0.1 nm, and allow for the analysis of extremely dense interference fringes and the measurement of even >1000 micron-thick films. The second approach is

based on InGaAs pixel arrays with 512 or 1024 pixels spread in a narrow spectral regime and along with narrow slits. In Figure 6a, the reflectance from a 1000 μm thick microscope glass, as recorded by a spectrometer based on first approach is illustrated. Clearly, the optical resolution of the spectrometer is high enough for the resolution of all dense interference fringes and the accurate measurement of even thicker films is possible. In Figure 6b, the reflectance from a 1000 μm thick microscope glass, as recorded by a spectrometer based on the second approach is illustrated. In the latter case ((b) probing strategy) the optical resolution of the spectrometer does not suffice for the resolution of the interference fringe at full amplitude.

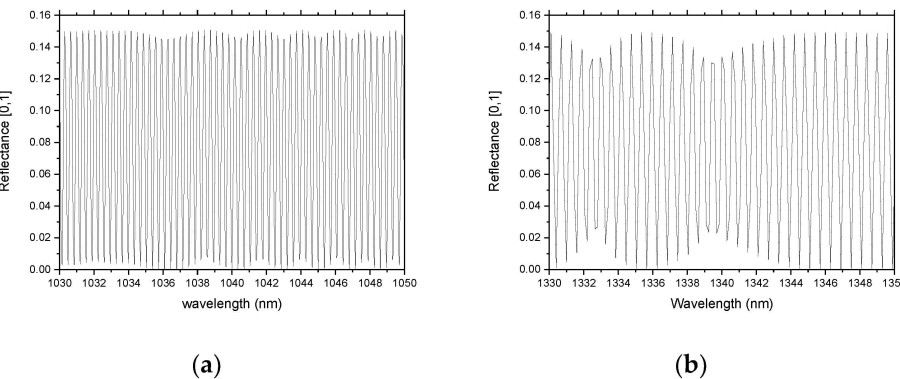

|     |     |
| :-: | :-: |
| (**a**) | (**b**) |

**Figure 6.** (**a**) Theoretical reflectance from a 1000 μm thick microscope glass using Si-CCD-based spectrometers with 3600 pixels accompanied with narrow entrance slits. (**b**) Reflectance from a 1000 μm thick microscope glass using InGaAs pixel arrays with 512 or 1024 pixels spread in a narrow spectral regime and along with narrow slits. Clearly, the resolution is quite satisfactory in case (**a**) but not in case (**b**).

In Figures 7 and 8, the thickness measurement of double-side polished Si wafer and microscope glass is demonstrated by employing a tool tuned to operate in the 900–1050 nm spectral range and equipped with 3648 pixels Si-CCD. In the case of thick and ultra-thick films, the spectral content of dense interference fringes can be also analyzed by appropriate fast Fourier algorithms, which provide ultra-fast measurement of film thickness with very limited difference from the thickness measurement with the Levenberg–Marquardt algorithm. The Levenberg–Marquardt algorithm [17] is a very fast and robust algorithm for large-scale, non-linear least-squares problems, and for that reason has been employed in fitting the reflection spectrum for ultra-thin, thin and thick films, while FFT can be employed in the measurement of thick films.

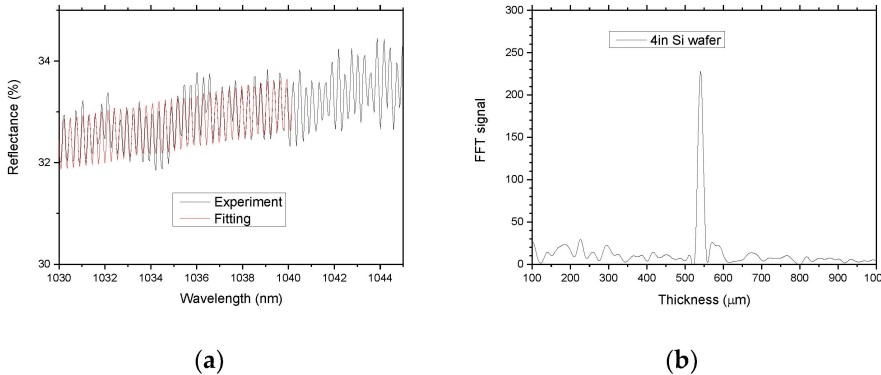

|     |     |
| :-: | :-: |
| (**a**) | (**b**) |

**Figure 7.** (**a**) Theoretical reflectance and fitted spectra in a narrow spectral regime from a 4-inch Si wafer. The spectrometer is equipped with Si-CCD with 3600 pixels accompanied by a narrow entrance slit. The fitting is performed by the Levenberg–Marquardt method. (**b**) FFT spectrum from the same reflectance spectrum.

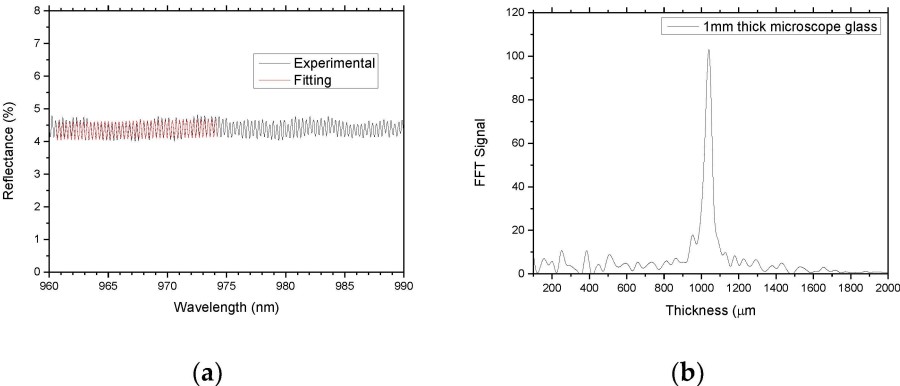

(**a**)                    (**b**)

**Figure 8.** (**a**) Theoretical reflectance and fitted spectra in a narrow spectral regime from a 1 mm thick microscope glass. The spectrometer is equipped with Si-CCD with 3600 pixels accompanied by a narrow entrance slit. The fitting is performed by the Levenberg–Marquardt method. (**b**) FFT spectrum from the same reflectance spectrum.

### 3.4. Characterization of Transparent Layers on Transparent Substrates

Another interesting and challenging application domain is that of transparent films applied on transparent substrates [18,19]. In those cases, the refractive index difference between the adjacent layers is limited and therefore the accurate thickness measurement presents a challenge.

In Figure 9, the successful measurement of film thicknesses of two mainstream photolithographic materials spin-coated on fused silica glasses is illustrated. In Figure 9a, the case of 1.1 µm thick AZ-5214 resist is shown. Clearly, the amplitude of the interference fringes is lower compared to the same film when applied on Si substrate, due to the limited refractive index difference between the resist and fused silica. Similarly, the successful measurement of thick SU-8 resist on microscope glass is shown in Figure 9b.

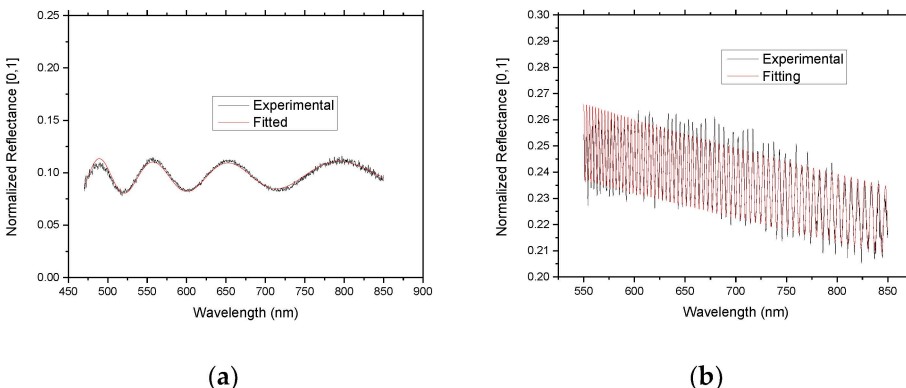

(**a**)                    (**b**)

**Figure 9.** (**a**) Theoretical reflectance and fitted spectra for AZ5214 photoresist on fused silica glass, AZ-5214 resist thickness 1.11 µm. (**b**) Theoretical reflectance and fitted spectra for SU-8 on microscope glass, SU-8 resist thickness 32.2 µm.

### 3.5. Multi-Layer Films

In several practical applications, the layer stack is not a single film over a substrate but rather a multi-layer on the substrate, with the multi-layer comprising layers with different optical properties [20,21]. In those cases, the requirement is the simultaneous measurement of the individual thicknesses, a rather challenging task especially when the adjacent layers exhibit refractive indices that do not differ widely.

In Figure 10, the characterization of a typical double-layer microelectronics application is illustrated. It comprises a Si substrate, thermal $SiO_2$ grown on the latter and a top thin $Si_3N_4$ layer deposited by LPCVD.

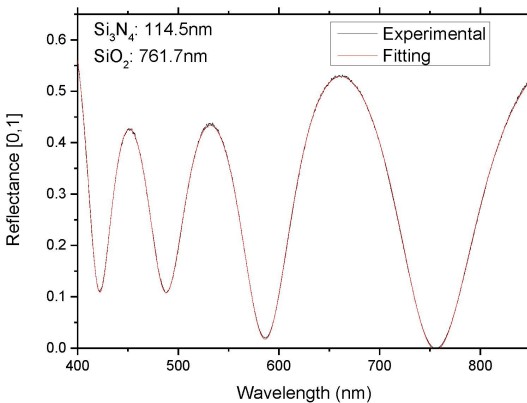

**Figure 10.** Experimental and fitted reflectance spectra for a double layer of Si$_3$N$_4$ and SiO$_2$ on Si substrate.

In Figure 11 the characterization of a rather standard layer stack for planar waveguides operating in visible range is illustrated. The layer stack consists of thick SiO$_2$ that acts as a bottom cladding layer, a thin Si$_3$N$_4$ layer as core of the waveguide and a patternable PMMA layer. Clearly, the matching of the experimental and fitted spectra is very good, and the thickness derived for each layer is in very good agreement with the thickness values of single films, as in the case of the SiO$_2$ layer, which was characterized before the deposition of silicon nitride.

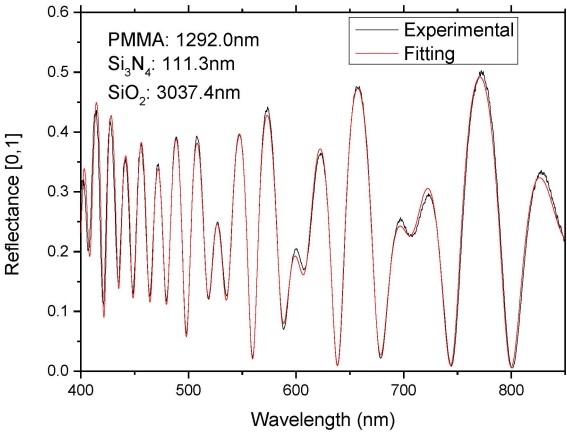

**Figure 11.** Experimental and fitted reflectance spectra for a tri-layer of PMMA on Si$_3$N$_4$ and SiO$_2$ on Si substrate, for planar waveguide applications.

Therefore, the reflectance spectroscopy methodology can be employed in the characterization of either multi-layer films and/or heterogeneous film. In Figure 12, such a case is demonstrated. In particular, in Figure 12a,b, the thickness maps of two transparent layers deposited on 6-inch quartz wafers are illustrated. In Figure 12a, the thickness map of the bottom layer (polyimide) deposited on the quartz wafer is illustrated along with the related color map. Similarly, in Figure 12b, the thickness map of the top layer (NdBR) is illustrated along with the related color map.

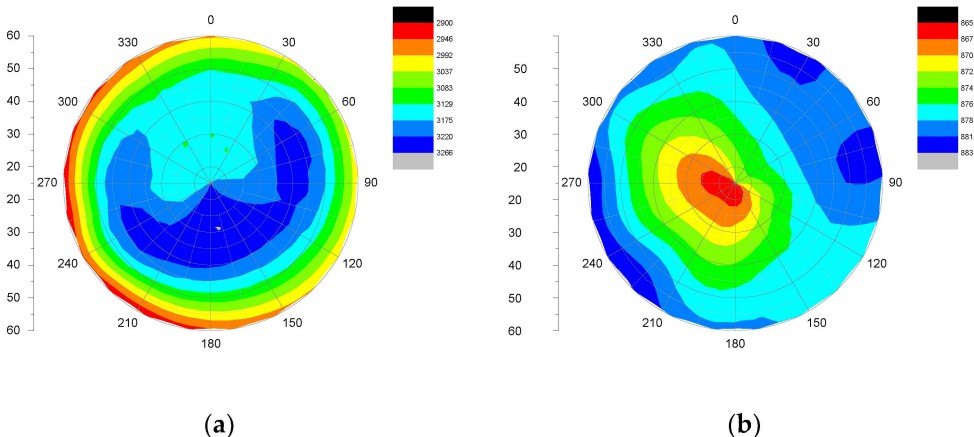

(**a**) (**b**)

**Figure 12.** Thickness maps of two transparent layers on a transparent substrate (NdBR/PI/Quartz). Both thicknesses (NdBR thickness, PI thickness) were measured simultaneously. (**a**) Thickness map of the transparent layer (PI) on the quartz substrate. (**b**) Thickness map of the top layer (NdBR).

*3.6. Kinetics of Chemical and Physical Chemical Processes and Monitoring of Transitions*

Here, we consider films that undergo changes as a function of time. For example, interaction of polymeric films with certain analytes, either in gas or in liquid phase, causes a film thickness change, either swelling or dissolution [22–25].

We present two quite important case studies: (a) the photoresist film thickness evolution when immersed in appropriate developers after it has been exposed (standard process in microlithography) and (b) the film thickness evolution during heating (usually at fixed rate as in the case of glass transition temperature ($T_g$) determination).

In Figure 13, the thickness evolution of a standard photoresist film (AR-N7520) immersed in the recommended developer is illustrated. Clearly, for a significant time period, the photoresist film dissolves smoothly and at a constant rate. Once the photoresist film thickness reaches a certain value (which is ca. 150 nm in Figure 13), the dissolution proceeds at higher rate.

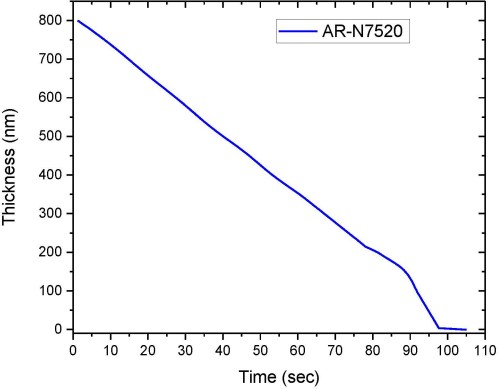

**Figure 13.** In situ monitoring of AR-N 7520 high-resolution negative resist monitoring in the appropriate developer.

In Figure 14, the thickness and refractive index of a poly (methyl methacrylate) (PMMA) film during heating is illustrated. Previously, the PMMA film was spin-coated on a Si wafer and then post-apply baked at 150 °C for the removal of any remaining casting solvent. The heating was performed at a constant temperature rate from 30 to 170 °C, while the temperature was recorded along with the reflectance spectrum. By fitting of the reflectance spectrum, it is possible to measure simultaneously film thickness and refractive index ($n$) as a function of temperature. Glass transition temperature ($T_g$) can be determined through the monitoring of different parameters as a function of temperature; proper choices

include dilatometric measurements (i.e., the monitoring of film dimensions as a function of temperature) and the monitoring of n as a function of temperature (dn/dT shows a jump at $T_g$). WLRS versions of the latter two types of measurements are possible simultaneously, as shown in Figure 14 for a PMMA film.

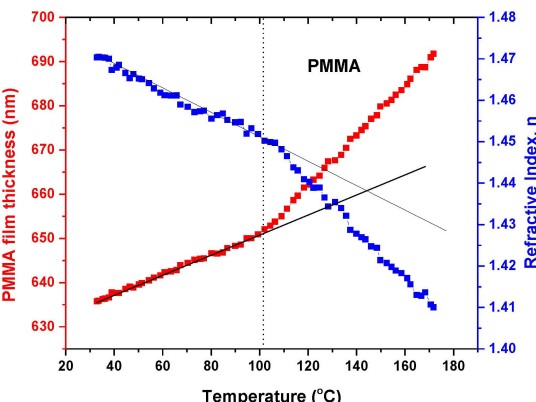

**Figure 14.** Thickness, d, and refractive index (at 633 nm wavelength) of a poly (methyl methacrylate) (PMMA) film during heating. The rate of change for both d and n changes following glass transition, which can be determined as located slightly above 100 °C.

### 3.7. In Situ Monitoring of Biomolecular Interactions

The monitoring of biomolecular interactions is essential for many bioanalytical applications, including diagnostics, drug screening, environmental monitoring and affinity sensing. Label-free detection is particularly attractive for monitoring of biomolecular interactions and reflectance spectroscopy can be proved a powerful tool in the real-time monitoring of biomolecular interactions [26–31]. However, the size of biomolecules is very small (with a thickness often in the 1 nm range) and the biomolecular adlayers that grow during biomolecular interaction are extremely thin and are not uniform (as a result of the fact that the quantum of growth is not a monolayer but a single molecule). Despite this, it was experimentally found possible to monitor affinity bioreactions, especially antibody–antigen reactions, in real-time, as it is depicted in Figure 15 where the binding of a goat anti-mouse IgG antibody onto mouse IgG molecules immobilized on a Si chip with different thicknesses of $SiO_2$ is monitored in real-time. Taking into account the variation of the baseline, i.e., the response variations prior to introduction of the reacting molecules solution, it was calculated that the minimum detectable mean biomolecular layer thickness was 0.1 nm. Thus, it was possible to monitor directly the binding of antigen onto immobilized specific antibody in cases where the antigen had high molecular weight, e.g., C-reactive protein (125 KDa) [29] and C3b complement component (185 KDa) [30]. On the other hand, for the detection of protein molecules of lower molecular weight, e.g., prostate specific antigen (~32 KDa), additional reaction steps implementing a second specific antibody was employed in order to provide the detection sensitivity necessary for the determination of the targeted analytes–antigens [31]. Similarly, for low molecular weight analytes (<1000 Da), the approach followed was the implementation of a competitive immunoassay format according to which the antigen is immobilized onto the sensing surface and competes with the antigen molecules in the sample for binding to the specific antibody [29]. Thus, the binding of the specific antibody (150 KDa) onto the sensor surface is what is actually monitored, and the signal is inversely proportional of the antigen concentration in the sample. In all cases, the implementation of additional reaction steps, e.g., by biotinylation of the specific antibody and use of streptavidin or by using an anti-species specific antibody (secondary antibody) could increase the signal corresponding to a fixed antigen concentration and thus help to reduce the assay time.

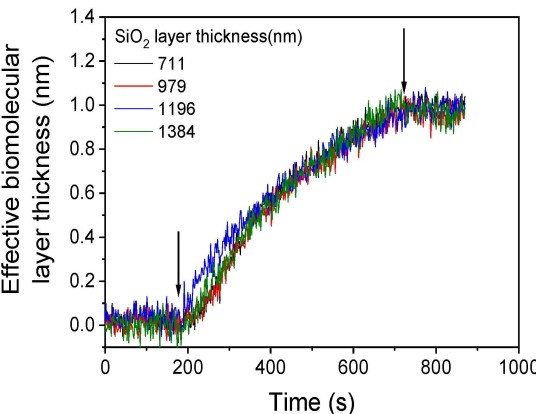

**Figure 15.** Real-time monitoring of goat anti-mouse IgG antibody to mouse IgG molecules immobilized onto Si chips with a SiO$_2$ layer of 711 (black line), 979 (red line), 1196 (blue line), 1384 nm (green line). The arrows indicate the time interval for which the anti-mouse IgG antibody solution was run over the chips.

*3.8. Refractive Index Measurement*

In addition to the application of WRLS in the measurement of film thickness, the reflectance spectroscopy method can be also applied for the simultaneous measurement of both film thickness and refractive index. For such determinations, the film and the substrate should be smooth and film thickness should be sufficiently thick (e.g., >100 nm for fitting in spectral range that starts at 400 nm). By applying the refractive index models, the refractive index dispersion can be calculated. In Figure 16, the results from the simultaneous calculation of film thickness and refractive index dispersion are illustrated. In particular, in Figure 16a, the experimental and fitted spectra of a SiO$_2$ film on Si is plotted, and in Figure 16b, the refractive index dispersion is illustrated.

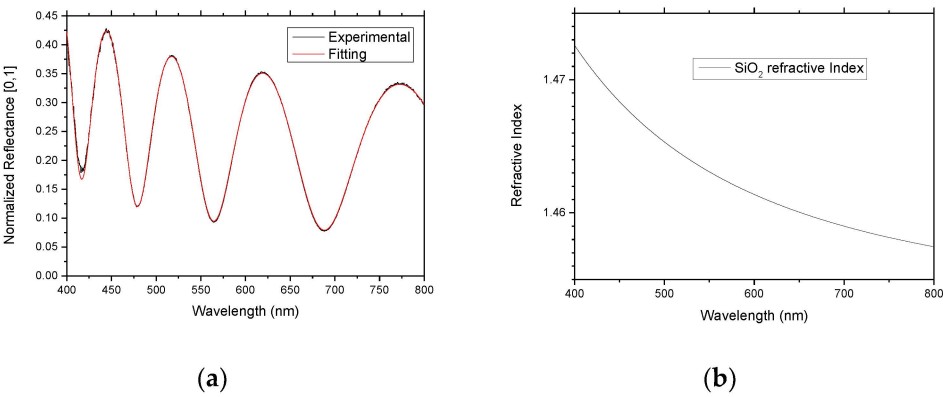

(**a**) (**b**)

**Figure 16.** (**a**) Experimental and fitted spectra for SiO$_2$/Si. (**b**) Calculated dispersion of refractive index of SiO$_2$ (Cauchy model).

**4. Conclusions**

White light reflectance spectroscopy is a fast, accurate, affordable, and non-destructive method for the characterization of thin and thick films, both under static and dynamic conditions. WLRS can be applied successfully for the measurement of thickness in the 1 nm to 1000 mm range (six orders of magnitude); at the same time, WLRS setups are small and easy to handle, and offer small spot size (through the use of microscope columns).

WLRS is found applicable to ultra-thin films (including 2D materials) and ultra-thick films, to transparent films and substrates and to multi-layer composite films. In addition, cases of successful real-time monitoring of film thickness changes include transitions (such as glass transition, probed as a function of temperature that increases at a fixed rate), biomolecular interactions and film dissolution of a negative resist.

**Author Contributions:** Conceptualization, K.G.B. and I.R.; methodology, I.R. and D.G.; validation, T.M., S.F. and G.Z., formal analysis, D.G., I.R. and investigation, D.G., I.R., T.M., S.F., G.Z., P.P. and K.G.B.; resources, I.R. and D.G.; data curation, D.G., I.R.; writing—original draft preparation, I.R. and K.G.B.; writing—review and editing, P.P., I.R. and K.G.B.; visualization, T.M., S.F. and G.Z.; supervision, I.R. and D.G.; project administration, I.R., D.G., P.P. and K.G.B. All authors have read and agreed to the published version of the manuscript.

**Funding:** This research has been co-financed by the European Regional Development Fund of the European Union and Greek national funds through the Operational Program Competitiveness, Entrepreneurship and Innovation, under the call RESEARCH–CREATE–INNOVATE (project code: T1EDK-00924 "ARCHIMEDES").

**Data Availability Statement:** The data supporting the findings of this study are available from the corresponding authors upon reasonable request.

**Conflicts of Interest:** The authors declare no conflict of interest.

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
