# Peer review of "Non-Destructive Characterization of Selected Types of Films and Other Layers via White Light Reflectance Spectroscopy (WLRS)"

_2673-8023, doi:10.3390/micro2030031_

Round 1

Reviewer 1 Report

This manuscript discusses the operational principles of White Light Reflectance Spectroscopy (WLRS) and presents a number of results on different applications of this non-destructive optical technique for determining film thicknesses in a wide range of values. Especially impressive are the results of measurements of extremely thin (1 nm) and extremely thick (1 mm) layers, characterisation of 2-D materials as well as of monitoring the dynamics of layer thicknesses in different chemical, physicochemical and biomolecular processes. I am convinced that the article will be of interest to a wide audience and in particular to the ridership of the journal micro. Therefore, I recommend publishing the paper with minor revision taking into account the following comments and suggestions:

1.     I suggest to reverse the order of the second sentence in the Introduction (lines 30 & 31): first “… produce …” and then “… accurately control…”

2.     The authors list several methodologies for the measurement of the thickness of coatings but discuss the advantages and disadvantages of only ellipsometry and reflectance spectroscopy. They should either discuss the others techniques as well or give a proper reason for not doing this.

3.     In line 56 please change “spot-side” to “spot-size”.

4.     In line 75 the ending of a sentence “…and illuminate it 90°” needs a revision.

5.     The curves in Figure 3. (a) cannot easily be distinguished, which makes the plot difficult to understand.

6.     Please briefly introduce the Levenberg-Marquart method and explain why it was selected to fit the experimental results presented in Figures 7 & 8.

7.     The sentence in lines 238-241 is not clear and should be revised.

8.     “Figure 11a” in line 243, “Figure 11b” in line 244 and “Figure 12” in line 260 should be changed to “Figure 12a”, “Figure 12b” and “Figure 13”, respectively.

9.     It is not quite clear what is the difference between the two plots (a and b) in Figure 12. This should be better explained in the text and figure caption.

10. Why is the PMMA layer getting thicker with increasing temperature in Figure 14? One could expect that with the removal of the solvent, the resist layer will become thinner.

Author Response

We would like to thanks the Reviewer for his/her comments that help us to prepare a more clear and informative manuscript

Manuscript ID: micro-1777356

Title: Non-destructive Characterization of Selected Types of Films and Other Layers via

White Light Reflectance Spectroscopy (WLRS)

Manuscript Type: Article

Reviewer 1

This manuscript discusses the operational principles of White Light Reflectance Spectroscopy (WLRS) and presents a number of results on different applications of this non-destructive optical technique for determining film thicknesses in a wide range of values. Especially impressive are the results of measurements of extremely thin (1 nm) and extremely thick (1 mm) layers, characterization of 2-D materials as well as of monitoring the dynamics of layer thicknesses in different chemical, physicochemical and biomolecular processes. I am convinced that the article will be of interest to a wide audience and in particular to the readership of the journal micro. Therefore, I recommend publishing the paper with minor revision taking into account the following comments and suggestions:

Comment-1. I suggest to reverse the order of the second sentence in the Introduction (lines 30 & 31):

first “… produce …” and then “… accurately control…”

Answer: we thank the Reviewer for the comment. The particular sentence was revised.

Comment-2. The authors list several methodologies for the measurement of the thickness of coatings but discuss the advantages and disadvantages of only ellipsometry and reflectance spectroscopy. They should either discuss the other techniques as well or give a proper reason for not doing this.

Answer: As suggested by the manuscript’s title, this is a work about the capabilities offered by a particular methodology (reflectance spectroscopy) that we know in substantial detail and we aim at sharing our understanding by presenting the pertinent essentials of a broad range of types examples.

Of course, one can write more about a number of methodologies that have similar characterization and/or monitoring aims but this would be somewhat superficial, if not part of a much longer article. It would be much better if articles similar to ours appear in this Journal about the possibilities offered by these other methodologies. We only make some somewhat detailed comment about ellipsometry because it is another optical reflectance methodology and it is somewhat related to the methodology probed herein.

Comment-3. In line 56 please change “spot-side” to “spot-size”.

Answer: Corrected

Comment-4. In line 75 the ending of a sentence “…and illuminate it 90°” needs a revision.

Answer: we thank the Reviewer for the comment. The particular sentence was revised.

Comment-5. The curves in Figure 3. (a) cannot easily be distinguished, which makes the plot difficult to understand.

Answer: The number of curves in figure 3a is indeed large and the differences at high wavelengths are small, especially in the case of ultra-thin films. For that reason, we have included figure 3b in which we present the same spectra for a narrow spectral regime (200-400nm) in order to show in detail the difference as regards reflectance.

Comment-6. Please briefly introduce the Levenberg-Marquart method and explain why it was selected to fit the experimental results presented in Figures 7 & 8.

Answer: In the methodology presented in this manuscript the user sets a thickness range and then an algorithm should search for the actual film thickness(es). In several cases this initial thickness range is quite broad and, hence, a fast algorithm should be employed. The latter need becomes more pronounced when more than one thicknesses should be measured (multi-layer case), or in cases of in-line thickness measurement (production environments). The Levenberg-Marquart algorithm is a very fast and robust algorithm for large scale nonlinear least-squares problems and for that reason has been employed in the fitting of the reflection spectrum for ultra-thin, thin and thick films, while FFT can be employed in the measurement of thick films. For the implementation of this method in the measurement of film thickness(es) a detailed study on the selection of the seed values has been performed; a relevant note has been added to the manuscript and one reference about the Levenberg-Marquart algorithm was added (reference 17).

Comment-7: The sentence in lines 238-241 is not clear and should be revised.

Answer: The sentence in question is the following: More generally, it follows that we can probe films that have dimensions fixed in time while in the normal to the surface direction the material is not the same either along the latter direction alone (case of multilayer composites) or at different substrate coordinates (case of inhomogeneous films).

We have tried to be precise as regards the types variations of material that can be probed best. Yet as the description is somewhat difficult to follow, we replaced the original sentence with the following more general one: “Therefore, the Reflectance Spectroscopy methodology can be employed in the characterization of either multilayer films and/or inhomogeneous films”

Comment-8. “Figure 11a” in line 243, “Figure 11b” in line 244 and “Figure 12” in line 260 should be changed to “Figure 12a”, “Figure 12b” and “Figure 13”, respectively.

Answer: Corrected

Comment-9. It is not quite clear what is the difference between the two plots (a and b) in Figure 12. This should be better explained in the text and figure caption.

Answer: We thank the Reviewer for the comment. These two plots refer to thickness maps of the two layers on the transparent substrate. The two transparent layers were PI (bottom layer) and NdBR (top layer). Related explanations were added both in the text and the figure caption

Comment-10. Why is the PMMA layer getting thicker with increasing temperature in Figure 14? One could expect that with the removal of the solvent, the resist layer will become thinner.

Answer: The solvent is gone when the film is post apply baked (‘for the removal of any remaining casting solvent’, as stated).  What we present is the monitoring of the solvent-free film during subsequent heating (in which case the film expands with heating). We now make clear that we monitor the film (as a function of temperature) after it was post apply baked by appropriately revising the related text “Previously the PMMA film was spin-coated on a Si wafer and then post apply baked at 150oC for the removal of any remaining casting solvent

Reviewer 2 Report

The topic of the article is quite original, and after reading the introduction, the reviewer expected to get acquainted with a small overview of the existing applications of the white light reflection method. But in the Methodology section, the measurement technique and the theoretical basis are described in detail. There is no mention of the objects of study and the procedure for converting formula (1) from the reflection signal to the layer thickness. Is there a ready-made mathematical apparatus for converting?

What is the measurement error as a consequence of subsequent thickness calculations?

Section 3.1 presents the measurement data of the reflection signal in ultra-thin films, the thickness of which was measured by another method. The authors showed that the method is applicable for such a case. But do the authors have data on their measurements of the thickness of ultra-thin films by the proposed method?

What advantage does white light reflection have over other methods, such as spectroscopic ellipsometer? Does the proposed method have any limitations in an application? Can it be used to estimate the film thicknesses of compounds for which the reflection indices are unknown?

Despite the comments presented, the present article is of great interest for estimating the thickness of coatings by a non-destructive method. The authors should go into more detail on the aims and objectives of the investigation.

Author Response

Dear editor,

we submit our feedback on the issues raised bt Reviewer-2

sincerely

Ioannis Raptis
